# Organic Matter and Associated Minerals on the Dwarf Planet Ceres

**Maria Cristina De Sanctis [1],*** and **Eleonora Ammannito [2]**

[1] Istituto di Astrofisica e Planetologia Spaziali, Istituto Nazionale di Astrofisica (INAF), 00133 Rome, Italy
[2] Agenzia Spaziale Italiana, Via del Politecnico, 00133 Roma, Italy; eleonora.ammannito@asi.it
* Correspondence: mariacristina.desanctis@inaf.it

**Abstract:** Ceres is the largest object in the main belt and it is also the most water-rich body in the inner solar system besides the Earth. The discoveries made by the Dawn Mission revealed that the composition of Ceres includes organic material, with a component of carbon globally present and also a high quantity of localized aliphatic organics in specific areas. The inferred mineralogy of Ceres indicates the long-term activity of a large body of liquid water that produced the alteration minerals discovered on its surface, including ammonia-bearing minerals. To explain the presence of ammonium in the phyllosilicates, Ceres must have accreted organic matter, ammonia, water and carbon present in the protoplanetary formation region. It is conceivable that Ceres may have also processed and transformed its own original organic matter that could have been modified by the pervasive hydrothermal alteration. The coexistence of phyllosilicates, magnetite, carbonates, salts, organics and a high carbon content point to rock–water alteration playing an important role in promoting widespread carbon occurrence.

**Keywords:** Ceres; asteroids; organics; aliphatics; solar system





## 1. Ceres: A Dwarf Planet in the Main Belt

The dwarf planet Ceres stands out among the asteroids because it has physical and chemical characteristics that are peculiar with respect to those of the typical main belt bodies. Ceres is the largest object in the main asteroid belt and also the most water-rich body in the inner solar system after Earth. Ceres comprises nearly one third of the mass of the asteroid belt (mean radius = 470 km; bulk density = 2162 kg m$^{-3}$ [1] and shares similar characteristics with icy satellites of the outer solar system [2,3] in terms of radius, density, and presence of abundant water ice (Figure 1). The Dawn Mission made a comprehensive study of this body, revealing unexpected characteristics that make it an interesting target in the search for life [3,4]. The recent observations of the Dawn Mission [1] show evidence for modern and even ongoing geological activity on Ceres [2,5,6], possibly driven by the presence of salty liquid below an ice-rich crust [2,6]. $H_2O$ and $OH$ molecules have been detected around Ceres [7,8] and optical instruments onboard the Dawn spacecraft observed evidence of recent brine-driven exposure of material onto Ceres' surface. The most striking evidence was discovered at Occator Crater [2,9,10] and at the young Ahuna Mons [5,11,12].

The data acquired by the VIR (visual and infrared) mapping spectrometer on the Dawn spacecraft [13] unambiguously detected local high concentrations of organic matter [14–16], while the presence of carbon, distributed globally, has been inferred combining the data of the VIR and GRaND (Gamma-Ray and Neutron Detector) spectrometers [17–19]. The presence of carbon was also suggested by ultraviolet (UV) spectral observations [20]. Models and data reveal that Ceres has sufficient water and silicates to have developed an ocean throughout its history, leading to a layered interior structure with a high degree of aqueous alteration [3] that is confirmed by the observed surface mineralogy [21]. Ceres was initially considered a "Relict Ocean World" [2] given several similarities with other ocean worlds in the outer solar system, but the paradigm from "relict" to "present" is

changing with the new discoveries and modelling performed in the last period. In fact, the present-day existence of brines and salty water seems to have been confirmed by different observations [2,6], bringing Ceres in the list of the ocean worlds and astrobiological targets. In fact, Ceres mineralogy that includes phyllosilicates, carbonates and organics, coupled with the geological activity, make Ceres a primary target for the exploration of bodies with high potential for astrobiology [3,4]. Being an ice- and organic-rich body, Ceres potentially represents the prototype of the objects that accreted into the terrestrial planets, bringing water and organics in the inner region of the early Solar System.

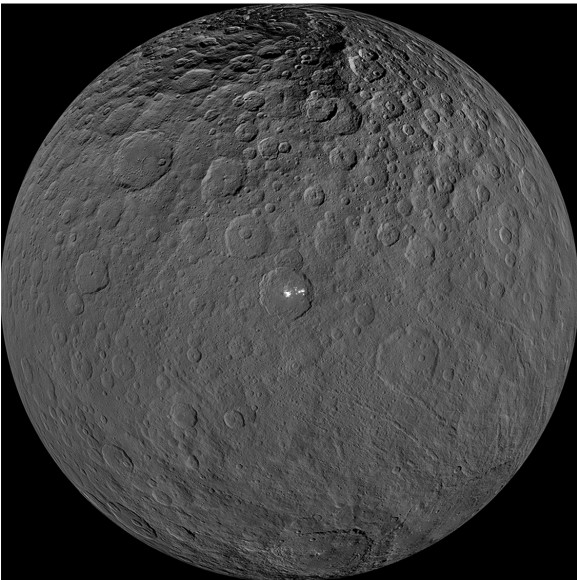

**Figure 1.** Image of Ceres' surface made by Dawn's framing cameras in clear filter. The brightest region on Ceres, called Cerealia Facula, is highlighted in Occator Crater in the center of this image. Vinalia Faculae, the set of secondary bright spots in the same crater, are located to the right of Cerealia Facula (Image credit: NASA/JPL-Caltech/UCLA/MPS/DLR/IDA).

## 2. Ceres Composition and Comparison with Asteroids and Meteorites

### 2.1. Ceres Composition

Ceres surface mineralogy has been inferred by ground based [22] and remote-sensing observations [15,17,21,23,24], consisting of a dark assemblage of different minerals, mostly resulting in aqueous alteration of silicates and carbon. The visible (VIS) and infrared average spectrum of Ceres shows clear bands at 2.7, 3.1, and 4 μm, attributed respectively to Mg-phyllosilicates, $NH_4$-phyllosilicates and Mg,Ca-carbonates (Figure 2) [23]. A broad band at about 3.3–3.5 μm is also present in the spectrum, attributed to the contribution of different minerals, primarily carbonate and ammoniated phyllosilicates, but also organics. In particular, aliphatic organics contribute to that signature in the organic rich-regions (hundred km sized areas close to Ernutet crater), where the absorption at 3.3–3.5 dominates the spectrum [10,14,15,25].

The spectrum also shows a large and shallow band at about 1 μm, likely associated with the presence of iron in the silicate [24]. Overall, the very low reflectance indicates the presence of dark and opaque phases that are difficult to identify unequivocally, given their featureless characteristics. Both carbon and magnetite can contribute to the dark material on Ceres and the retrieved abundances are not firmly estimated using only the VIS and IR spectra [17,19,25]. Even if ambiguity remains about the nature of the dark phase, the other minerals clearly identified in the Ceres average spectrum are aqueous alteration products, and imply a longstanding interaction with liquid water.

The elemental data acquired by GRaND on the Dawn mission [18] indicate that the globally averaged abundances of carbon (C) are about 8–14 wt.% and iron (Fe) are about 15–17 wt.%. Ceres' global reflectance spectrum and elemental abundances are best

fitted by a mixture of carbonaceous-chondrite-like materials (40–70 wt.%), insoluble organic matter (IOM) or amorphous carbon (10 wt.%), magnetite (3–8 wt.%), serpentine (10–25 wt.%), carbonates (4–12 wt.%), and $NH_4$-bearing phyllosilicates (1–11 wt.%), taking into account the GRaND instrument data to constrain the unmixing models. It must be recalled that carbonaceous-chondrites (CC) are, on average, composed by mixtures of unaltered and altered phases, including phyllosilicates and carbon, the latest in the forms of carbonates and organic matter. The carbonates include both calcite ($CaCO_3$) and dolomite ($CaMg(CO_3)_2$), while the organic component is mostly in the form of insoluble kerogen-like macromolecules, combined with less aliphatic compounds [26]. As a consequence, including carbonaceous-chondrites as the main component of the Ceres-like mixture implies an even more abundant quantity of carbon and organics as described by [17].

Some attempts in reproducing the average spectrum of Ceres in the laboratory have been made [27] using mixtures of phyllosilicates, with and without ammonium, carbonates and dark phases. The results suggest that the mixtures are able to reproduce the average spectrum of Ceres, even if some minor discrepancies are still to be solved.

### 2.2. Similarities and Differences with Carbonaceous Chondrite (CC) Meteorites

Ceres does not have an asteroidal family [28] and has not been directly sampled. Thus, we do not have meteorites from Ceres. Nevertheless, carbonaceous chondrite (CC) meteorites, especially CM and CI groups, have been considered the closest analogs for Ceres (e.g., [29–31]). Laboratory spectra of CM and CI carbonaceous chondrites measured under anhydrous conditions [31,32] show the same prominent 2.7 µm OH absorption band as the Ceres average spectrum, typical of Mg-phyllosilicates (Figure 2). This characteristic is similar to those of highly altered CM [33] and CI chondrites [34]. Also, the measurements of other CC under an asteroidal environment (high vacuum and temperature) show similarities with the Ceres spectrum (Figure 2) [31]. However, the spectra of these chondrites are not a perfect match with Ceres at other wavelengths and, even if the average spectrum of Ceres is broadly similar to that of CM chondrites, specific spectral bands indicate variations in the mineral mixture composing the surface. The most remarkable difference is that CM and CI chondrites lack Ceres' features at 3.05–3.1 µm (Figure 2b), attributed to the presence of NH4 in the phyllosilicates [23,35], as demonstrated by spectral measurements on ammoniated-phyllosilicates in laboratory [27,36–38]. Ammoniated clays have not been identified in carbonaceous chondrites, but $NH_3$ is a component of organics in CM/CI chondrites [39].

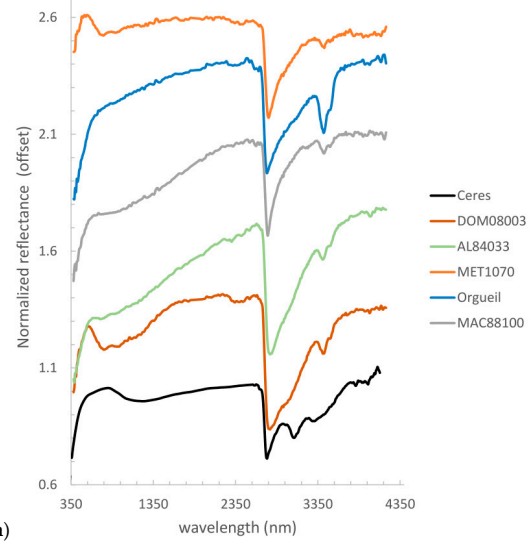

(**a**)

**Figure 2.** *Cont.*

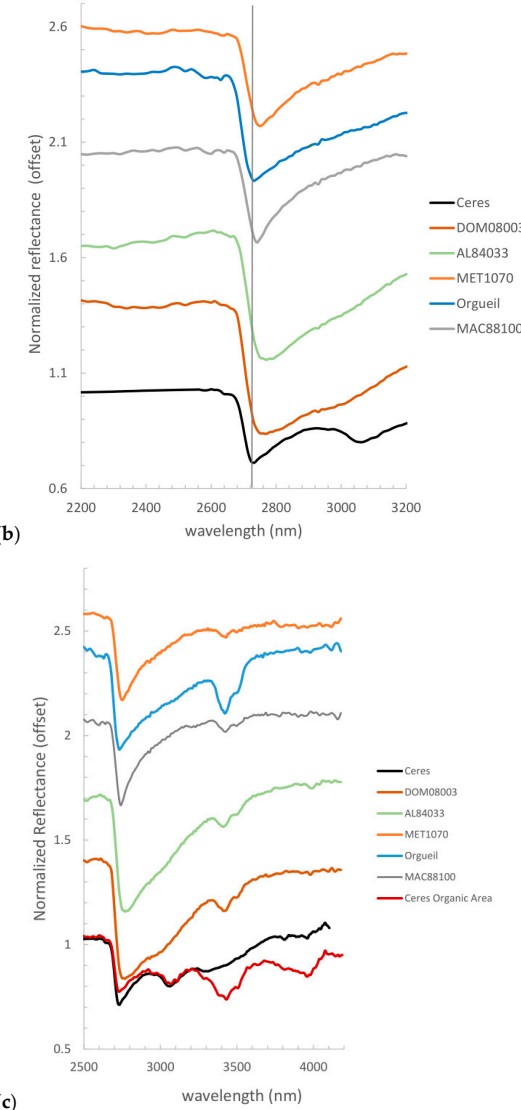

**Figure 2.** (**a**) Comparison between the spectra of Ceres and heated carbonaceous chondrites (Ceres from [24]; Heated carbonaceous chondrite (CC) from [31], SShade database). (**b**) Zoom in the region of the phyllosilicate absorptions. Comparison between the spectra of Ceres and heated carbonaceous chondrites (Ceres from [24]; heated CC from [31], SShade database). (**c**) Comparison between the spectra of Ceres (average-black spectrum, organic-rich region-red spectrum) and heated carbonaceous chondrites chondrites (Ceres from [24]; heated CC from [31], SShade database).

Also, the elemental composition derived by the Dawn/GRaND data implies that Ceres is rich in C and poor in Fe relative to CCs [18]. Thus, some clear differences are present between Ceres and CC.

The organic content of CC is quite high in relation to total carbon and it is complex in composition: most of the organic carbon in CC (about 70–99% of the total) is present as a complex and insoluble macromolecular material (IOM). In addition to carbon, the IOM also contains elements such as H, O, N, and S, with elemental composition that varies between different classes and subclasses. The soluble part (SOM) is minor. The IOM organic component of CC is variable in terms of relative abundance and composition [40]. The CO, CV, and CK have almost only insoluble kerogen-like material, while in the CI, CM, and CR classes there are also numerous soluble organic compounds. Moreover, there are differences in the extracted organic materials between meteorites of the same class [41,42].

The formation of the insoluble organics in CC, its origin, and its relationship with the soluble organic material is a matter of debate. There are some indications, such as the fact that carbonaceous material is organized in well-defined macromolecular structures, suggesting the origin of a portion of the IOM pre-dated parent-body processes. Nevertheless, some analyses (i.e., isotopic and pyrolytic) indicate that at least a quantity of insoluble material was possibly modified during parent-body processes. This alteration caused the graphitization and chemical oxidation, with the likely release of soluble organic compounds.

Soluble organics in CC meteorites include a various and complex suite of compounds that range from large species (i.e., polycyclic aromatic hydrocarbons (PAHs)) to small molecules (i.e., methane). Amino acids compounds are an important part of organics and are found in some specific CC meteorites, such as CI, CM, CR, and the ungrouped Tagish Lake [41–43] meteorite. It is interesting to note that there is a dependence of amino acid with alteration degree of the meteorite: the relative quantity of amino acids declines with a growing degree of alteration in the CC, indicating that aqueous alteration processes has an influence on the organic matter [44]. Moreover, also the proportion of IOM in CM chondrites diminishes with increasing alteration while carbonate abundance increases [45].

In the case of Ceres, the retrieved quantity of carbonate is larger while the organics are less abundant with respect to CC, supporting the idea that Ceres underwent an advanced alteration.

### 2.3. Similarities and Differences with C Asteroids

Ceres is the largest body of the main belt, classified as C-type based on its spectral broad characteristics. However, its spectrum is very uncommon and there are only a couple of other asteroids (10 Hygiea and 324 Bamberga) showing a band at 3.06 μm similar to the Ceres' one [46]. Like Ceres, these asteroids show a 3 μm feature with a band center of $3.05 \pm 0.01$ μm, overlaid on a wider absorption from ~2.8 to 3.7 μm. Other C-type asteroids have a much shallower and broader 3 μm band, quite different from the Ceres' one. Both 10 Hygiea (444 Km in diameter) and 324 Bamberga (230 Km in diameter) are large asteroids. The thermal history and alteration degree of C-type asteroids can be related to their size and the presence of the 3.06 μm could be an indication of advanced degrees of alteration. However, there are also other large C-type asteroids, such as 52 Europa, 31 Euphrosyne, and 451 Patientia not showing spectral similarities to Ceres [46]. Thus, the simple connection of the size of a C-type asteroid with its degree of alteration does not fully work and the evolutionary paths of the C-type asteroids must be more complex.

Comparing Ceres with small C-type asteroids is not easy, due to the low signal-to-noise of spectra of small and dark bodies. However, the recent data collected on asteroids Bennu and Ryugu [47,48] permit a direct comparison. Both the two small asteroids, although they clearly show the presence of phyllosilicates with spectral signatures at about 2.7 μm, do not show evidence for bands at 3.06 μm. It is likely the parent bodies of those two small asteroids did not undergo an evolutionary path similar to Ceres or the original compositions were different from that of Ceres. It must be also considered that the present thermo-physical environment of those two asteroids is very different with respect to the Ceres' environment, being two Near Earth Asteroids and having suffered catastrophic collisions [49,50] that strongly influenced their present state.

Nevertheless, small bright veins have been identified on Bennu, with an infrared absorption feature around 3.4 μm [51], interpreted as the signature of carbonates, similar to the carbonates found in aqueously altered carbonaceous chondrite meteorites.

Several main-belt asteroids display changes in the reflectance between 3.2 and 3.6 μm, indicating the potential presence of a band, but the presence of the aliphatic organic features at 3.4 μm were never clearly detected among them. A 3.4 μm band in the spectra of (24) Themis were ascribed originally to aliphatic organics and ice [52,53]. The recent observations by Osirix–Rex of asteroid Bennu [54] show an absorption band between 3.2 and 3.6 μm that broadly resembles those of Themis, Ceres, and other main-belt asteroids

(Figure 3). This feature has been attributed to the presence of organic. However, most spectra have an absorption shape that matches a mixture of carbonates and organics, thus the contribution of the two species is difficult to disentangle [54]. Furthermore, the presence of bright carbonates veins have been identified on the surface of Bennu, suggesting that the 3.4 µm band can be also the expression of the carbonates [51]. The comparison of the Ceres spectra, both of the average spectrum and an example of the organic rich region, with the Bunnu spectra show several differences (Figure 3) but also some similarities. The two bodies share a strong phyllosilicate band and a band at about 3.4 µm. However, the overall shapes of the spectra are dissimilar, both in the visible and the infrared ranges.

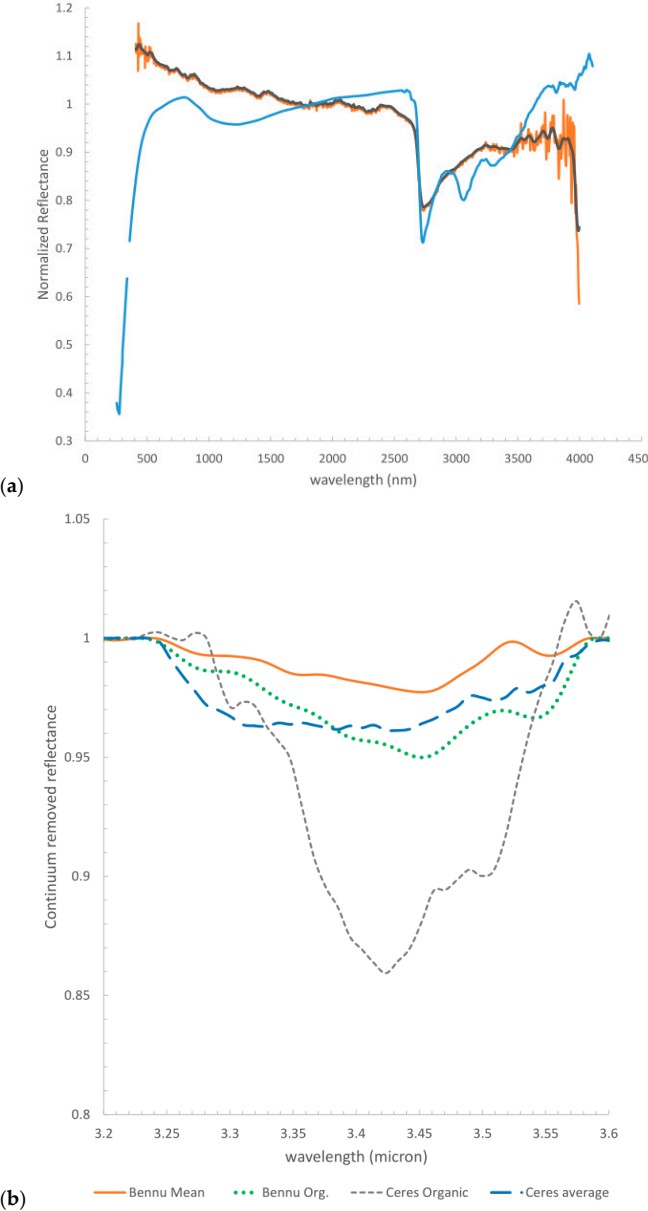

(**a**)

(**b**)

**Figure 3.** (**a**) Comparison between the average spectrum of Ceres (blue) and Bennu (not smoothed data in orange; smoothed data in grey). The spectra have been normalized at 2000 nm. Bennu spectra are from [55], Ceres spectrum from [24]. (**b**) Comparison between Ceres (blue-average spectrum, gray organic-rich average spectrum) and Bennu (average in orange; organic-rich in green). The spectra have been removed by the continuum in the range 3.25–3.6 µm. Bennu spectra are from [52,55]. Ceres spectra from [15,24].

Looking at the average Ceres and Bennu absorption, extending from 3.2 to 3.6 μm (Figure 3b), we recognize that Ceres has a stronger band with respect to Bennu. Also the shape of the band is different, with Ceres showing a broader band with a sort of minimum at about 3.3 μm (this feature can be associated also with the presence of ammonium in the phyllosilicate).

The comparison between the spectra of the organic-rich region of Ceres and that of Bennu shows different shape and depth. Ceres has a stronger band with clear minima at about 3.51 and 3.42 μm (see also [25]), while the minima of the organic band at Bennu are at 3.55 and 3.46 μm. The differences between the bands suggest that the mineralogy of the bodies is different and also the organics that characterize the surfaces of Bennu and Ceres have different composition.

The reason why Ceres is so dissimilar with respect other large bodies in the main belt is yet to be understood and could be related to a different formation region, further away and colder, with respect the Ceres' present position [23].

## 3. Distributed Organics over the Surface

Organic matter can be detected in the VIR spectral range (0.5–5.0 μm) thanks to the vibration absorption features and their overtones and combinations. The identification of the organic matter with reflectance spectroscopy, however, is particularly challenging due to the overlapping of the absorptions of functional groups from organic and inorganic compounds as well as the fact that non-vibrational modes of organic matter are IR active (i.e., present in the IR spectrum). Particularly relevant for the VIR spectral range are the absorption features associated with the C–H bonds in the organic matter. Aliphatic C–H bonds in CH2 and CH3 groups give rise to three distinct absorption bands near 3.3–3.6 μm while aromatic C–H absorption bands occur at slightly shorter wavelengths such as 3.1–3.3 μm.

A ubiquitous C–H band of aromatic carbons is not clearly observed in Ceres global spectra; we cannot, however, rule out the possibility of general presence of aromatic carbons. The signature of the C–H stretching might be indeed difficult to observe due to a very low absorption cross-section. Moreover, aromatic carbon might also exist without having a hydrogen bond (i.e., without a C–H stretching band). The IR spectra of chondritic IOM [55] and of several organic-rich shales [56], do not show the C–H stretching feature, although the carbon present in these organic material is found, for at least half of its abundance, in aromatic compounds [40]. Conversely, mature terrestrial organic matter, such as kerogens or coals, show absorption bands at 3.27 μm, due to a high abundance of aromatic mixtures with 'free' C–H aromatic bonds [56,57]. It has been proposed that the carbon aromatic rings in the IOM in CC might contain abundant methyl and aliphatic chains or alcohol groups attached to the aromatic ring. This structure results in a low abundance of free C–H aromatic bonds [40]. By contrast, mature terrestrial material contains less cross-linking. Therefore, the lack of a clear C–H aromatic stretching feature in the spectra of Ceres do not exclude the presence of aromatic organics but instead can give some hints to the cross-linking level of the carbon structure.

The detection of the C–H band of aliphatic chains is similarly critical. Although a weak 3.4 μm band could be an indication of the presence on the surface of widespread aliphatic chains [23], in the global spectrum of Ceres measured by the VIR spectrometer there is no clear evidence of the presence of such compounds. The aliphatic organics are known to be susceptible to C–H bond damage by ultraviolet radiation and energetic proton flux with a clear effect of lowering the H/C ratio [58,59] and the progressive destruction of the 3.4 μm absorption. Ceres's regional spectra show an ultraviolet minimum at 0.21 μm, interpreted as due to low H/C ratio organics [54]. Therefore, the presence of pervasive aliphatic matter also cannot be ruled out from the measurements by VIR.

It was shown by [17] that the global spectrum of Ceres can accommodate up to 60 vol% of carbonaceous chondrite material which would imply the possibility of meteoritic-like organic matter present on the surface without showing a clear feature associated with

organic matter in the 3.3–3.6 μm spectral range. The estimation by [17] would lead to roughly 20 wt% of carbon on the surface of Ceres which is more than five times higher than in carbonaceous chondrites. In carbonaceous chondrite, carbon is mainly present in carbonates and organic matter with the ratio between these two components related with the amount of aqueous alteration experienced by the meteorite [45].

## 4. Localized Organic Material

Despite the lack of a clear global feature associated with organic matter, near Ernutet crater, VIR detected a strong feature associated with the presence of aliphatic organics. The spectral feature in VIR spectra spans from 3.3 to 3.6 μm and is centered at 3.4, and it has been ascribed to overlapping bands of carbonates and aliphatic carbons. The feature has an asymmetric shape and is the combination of smaller features from 3.38 to 3.39 μm, from 3.40 to 3.42 μm, and from 3.49 to 3.50 μm [14]. The shape of this feature is also very akin to the 3.4 μm band observed in IOM of carbonaceous chondrites. The shape of the organic absorption band detected on Ceres shows noticeable matches with the organic bands of terrestrial hydrocarbons, like asphaltite and kerite [57]. Spectral modelling indicated an abundance up to ~9 vol% of organics, using as end-members terrestrial kerite-like materials [14]. However, more recently, it has been proposed that if the Ceres' organics are more similar to insoluble organic matter found in CC meteorites rather than terrestrial hydrocarbons, then the abundance inferred from spectral modelling may be locally higher, up to several tens of vol% [16]. In general, the values obtained for the abundance of organic matter are strongly dependent on the end-members used for the spectral modeling and also a change in the degree of aromaticity can affect the results [15].

The organic-rich area is part of the mid-latitude Coniraya quadrangle (Ac-2) in the northern hemisphere of Ceres [60]. The aliphatic organics have been observed as being mixed with other compounds, which are ubiquitous on the Ceres surface, but some materials, such as NH4-bearing minerals and/or carbonates, display an enrichment in the same region where the aliphatic organics were found [14,60].

However, there are small areas rich in organic material showing additional absorption bands. These areas display a strong absorption at 3.1 μm (Figure 4), indicating a high abundance of ammoniated-phyllosilicate. Moreover, an absorption at about 2.99–3 μm seems to be present. Even if the identification of the additional peak at 2.99 μm remains uncertain, it is likely that the 2.99 μm spectral feature is related with the N–H bond, either as an unidentified ammoniated-mineral or as organic amine compounds [15].

The organic rich regions identified on Ceres have a quite complex mineralogy, with the co-presence of sodium carbonates and clay-ammoniated compounds, often in excess with respect to the background (i.e., average Ceres). Therefore, an exogenous source for the OM would very likely require the delivery in the same event of the extra carbonate and ammonium found together with the organics. Both carbonate and ammonium-bearing materials, however, are not common on other bodies of the solar system therefore, it is very unlikely that such an unusual body could have impacted Ceres in recent times. For this reason, an endogenous source of the OM is more realistic when the general context of the OM rich-region is considered. In particular, aspects to consider are: the mineralogical composition of the region, the high retrieved abundances of organics in specific spots, and the observation of some likely extruded materials, even if they are not directly linked with the region of Ernutet. The bright material in the Occator crater, with high Na-carbonates signatures [9] and the water ice [61,62] and the cryovolcanism [5] have been explained by activity in Ceres' sub-surface. Thus, it is conceivable that the organic matter localized around the Ernutet crater can also be due to endogenous sources.

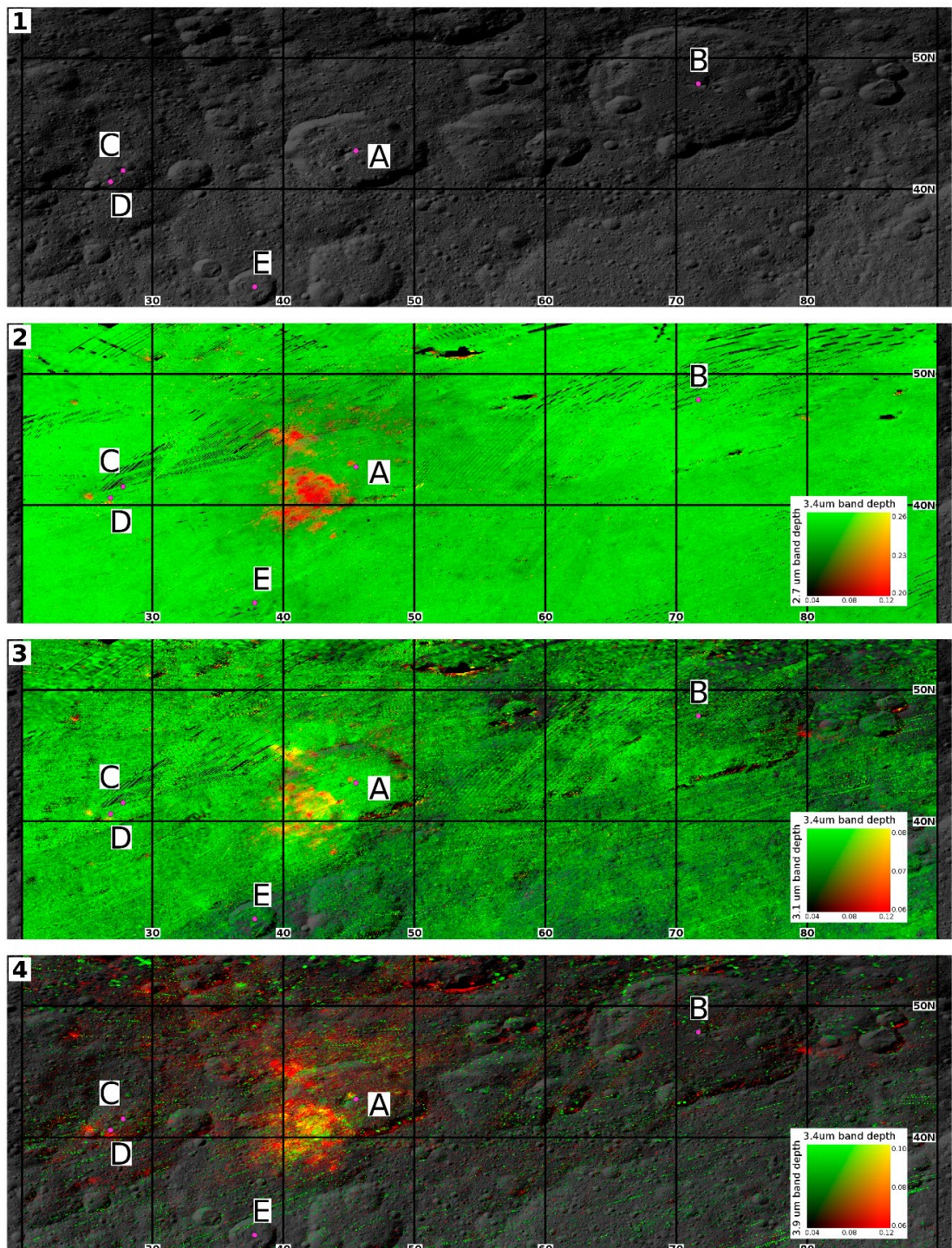

**Figure 4.** Interaction between the 3.4 μm band (organics) with other mineral phases identified on Ceres (i.e., phyllosilicates (2.7 μm band), ammoniated phases (3.06 μm band), carbonates (3.9 μm band)) in the Ernutet region. 1: clear filter mosaic; 2: abundances of organics in red and phyllosilicates in green; 3: abundances of organics in red and ammoniated phases in green; 4: abundances of organics in red and carbonates in green. Ranges are 0.20–0.26 for 2.7 μm band depth, 0.06–0.08 for 3.1 μm band depth, 0.04–0.12 for 3.4 μm band depth, 0.06–0.10 for 3.9 μm band depth. Morphological features in the figure are: A: Ernutet crater, B: Omonga crater, C: Hakumyi crater, D: Sukkot Labes, E: Liber crater.

## 5. Overview of Ceres Mineralogy and Presence of Organics

We have seen that organics and carbon are present on Ceres both at regional scale, such as the Ernutet region, and at global scale, as inferred by the modelling of the average spectrum and the elemental data [17]. The predominant signature in the spectrum of the Ceres organic-rich regions is the 3.4 μm band. This band has a shape and peak position consistent with the presence of a large quantity of organic material [14,16]. The most recent laboratory experiments suggest that about 20 wt% of OM in the form of both aliphatic and aromatic carbons can match Ceres spectra of the organic-rich areas [25]. It must be recalled that the analysis of the localized patches of organic material indicates that these organic-rich areas are associated with other minerals resulting from hydrothermal activity, such as sodium carbonate and ammonium clays, suggesting a common origin or combined evolution for these species. The spatial concentration of the clear signature of aliphatic organics on Ceres can be attributed to geologically recent exposure of those materials. It is known that aliphatic organics easily degrades when exposed onto the planetary surface by solar irradiation and heating. With irradiation, aliphatic carbon breaks down to graphitized or amorphous carbon, and these phases are thought to be everywhere on Ceres [17,20]. To preserve a clear 3.4 μm organic feature on Ceres, most of this material could have been exposed in geological "recent" time to preserve it from the degradation occurring in the space environment.

Ceres' surface has areas rich in carbonates and salts, that are the expression of recent, or even current, geological activity involving ascending hydrated material. The most remarkable are the faculae located in a crater named Occator. The spectrum of these faculae shows a strong absorption band in the 3 μm region that has been interpreted as the superimposition of several absorptions from different components: large amounts of Na-carbonate (50–80%), ammonium chloride, sodium chloride, and hydration. However, organic material can partially contribute to this band and the spectral unmixing models do not exclude such a contribution [2], but the spectral fitting results improve including the organic component in the mixture. Moreover, the visible part of the spectrum of the faculae has an unusual red (positive) spectral slope, similar to the visible spectral slope of the organic rich areas in Ernutet. The visible spectral redness has been considered as an indication of the presence of organic material, as suggested by the spectra of organic matter acquired in the laboratories [57]. Thus, there are several indications for a presence of organics in the bright faculae, including the recent exposure onto the surface.

The mechanism of formation of the material comprising the bright faculae involves liquid salty-water or brine ascending through cracks from a sub-surficial chamber. The inferred composition of the material constituting the faculae is similar to the composition of the Enceladus' plume, where sodium salts (NaCl, $NaHCO_3$ and/or $Na_2CO_3$) have been detected [63]. The plume also contains ice grains, simple organics and complex macromolecular organic material, originating from the subsurface ocean [64]. The similarities between the minerals in the Enceladus' plume and in the faculae suggest a Cerean subsurface reservoir where organic material can be present.

## 6. Discussion and Conclusions

The data acquired by the Dawn mission revealed that Ceres' composition includes organic material. There is the evidence of a large amount of localized aliphatic organics in specific areas and the presence of carbon has been inferred to be present globally.

From the outcome of the spectral modelling of the VIR instrument data from localized organic-rich regions, it seems that the estimated concentration of aliphatic carbons [15,16] is higher than the concentration of aliphatic organics normally found in CC meteorites [6] and the most recent laboratory measurements confirmed that the spectra of the organic-rich areas can be matched by mixtures with a large amount (about 20% weight) of aliphatic material [25]. Moreover, it has been derived that on the surface of Ceres' there might be up to 20 wt% of carbon, which is more than five times higher than the corresponding value for CC meteorites [17].

The C–H bonds in aliphatic organics are damaged by space-weathering. On the IR spectrum, the effect is a progressive suppression of the 3.4 μm absorption feature. Consequently, such absorption feature becomes difficult to detect especially when mixed with the carbonates [58]. Since a low H/C ratio in organics has been inferred for Ceres from a strong minimum in the spectrum at 2.21 μm [20], it is conceivable that organics are globally widespread on Ceres, but are not easily identifiable. Also, elemental data show that the regolith on the surface of Ceres could contain high concentrations of C embedded into carbonates and organics [17,19].

Similarly to other primitive bodies in the main belt, Ceres must have accreted organic matter as well as ammonia and carbon during its evolution. This was likely to happen since all these elements and compounds are present in the protoplanetary disk. The presence on the surface of Ceres of ammonium embedded into the phyllosilicates [21,23] is an indication that at least ammonia was accreted during the evolution of the dwarf planet and the pervasive presence of carbonates [23,65] points to the occurrence of the same mechanism for carbon. It seems logical, then, that the same should have happened for the organic matter. It is probable that the organic matter originally accreted was similar to meteoritic IOM. It is likely that, within Ceres there were the conditions for further processing such as the Fischer–Tropsch-type (FTT) reactions to produce hydrocarbons from carbon monoxide [15]. However, also prebiotic reactions based on hydrogen cyanide (e.g., [66]), formamide chemistry and/or formaldehyde chemistry [67,68] may have taken place. The activation of surface catalyzed reactions forming on phyllosilicates including the reduction of carbon dioxide to hydrocarbons was also possible [69].

In addition to that, another possible source of the organic matter found on the surface of Ceres is the material accreted from impacts. In the case of Ceres, most of the impact material would be represented by ordinary and carbonaceous chondrites, respectively representing S-Type and C-Type asteroids in the main belt [17]. The organic matter eventually delivered by impacts, however, would be similar to the one embedded by Ceres during its accretion with the difference of the level of processing due to the Cerean hydrothermal environment.

Experimental studies have shown that the hydrothermal alteration processes can be very efficient to rapidly produce several types of molecule such as insoluble materials starting from simple interstellar and/or protosolar molecules. The dissociation of molecules and the release of $CO_2$ as result of the hydrothermal alteration might have favored the formation of carbonates which, indeed, have on Ceres a concentration higher than in CC meteorites. This possibility is reinforced by the correlation observed in the organic rich terrains around Ernutet between the abundances of carbonates and OM [60]. In CC meteorites, it has been observed that the abundance of carbonates is directly related with the level of hydrothermal alteration of the parent body with high abundance of carbonates being typical of more altered meteorites. This is likely a consequence of the loss of their unstable fragments into the gas phase ($CO_2$, CO, $CH_4$, $NH_3$). This process might have been more efficient on Ceres rather than on chondrites. During the internal partial differentiation phase of Ceres, the most volatile compounds (i.e., organic matter and hydrated minerals) found themselves together in the outer layer crust. This would have allowed a higher rate of chemical reactions in the upper layer crust during the pervasive hydrothermal alteration. This line of events could explain the presence of organic matter in the upper crust of Ceres.

The present Ceres mineralogy as well as the high abundance of water ice indicate that during its formation Ceres must have accreted the organic matter, ammonia, water, and carbon that are present in the protoplanetary formation region. Given the original composition, Ceres may have also processed its own organic matter, that could have been modified by the pervasive and longstanding hydrothermal alteration on Ceres. In this scenario, Ceres represent a unique object, a sort of natural laboratory where likely the original molecules and elements had mobility and energy available to initiate chemical reactions typical of prebiotic chemistry. A better understanding of Ceres evolutionary path will provide new insights into the earliest phases of processes that eventually led to the

emergence of life in the inner solar system. To this end, in the near future a step forward is needed in Ceres exploration with instruments devoted to the investigation of organic chemistry during the early phases of the evolution of Ceres.

**Author Contributions:** All the authors have contributed to the manuscript. All authors have read and agreed to the published version of the manuscript.

**Funding:** This work has been supported by the Italian Space Agency (ASI-Italy). Grant: ASI I/004/12/2.

**Institutional Review Board Statement:** Not applicable.

**Informed Consent Statement:** Not applicable.

**Data Availability Statement:** Dawn data are archived in NASA's Planetary Data System; VIR spectral data may be obtained at http://sbn.psi.edu/pds/resource/dwncvir.html.

**Acknowledgments:** The authors thank S. Potin for providing the CC data from the SShade database (Schmitt, Bernard; Bollard, Philippe; Albert, Damien; Garenne, Alexandre; Gorbacheva, Maria; Bonal, Lydie; Volcke, Pierre, and the SSHADE partner's consortium (2018). SSHADE: "Solid Spectroscopy Hosting Architecture of Databases and Expertise" and its databases. OSUG Data Center. Service/Database Infrastructure, doi:10.26302/SSHADE). The authors thanks H.H. Kaplan and A. A. Simon for providing the data of Bennu.

**Conflicts of Interest:** The authors declare no conflict of interest.

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
