# Peer review of "Organic Matter and Associated Minerals on the Dwarf Planet Ceres"

_minerals, doi:10.3390/min11080799_

Round 1

Reviewer 1 Report

General aspects

The manuscript discusses the observations and possible origin of some organic components and other minerals identified at specific areas of the dwarf planet Ceres. It presents new results, important for the readers. The structure of the manuscript is good, its language is very good, the illustrations are relevant, the cited literature is also relevant. There are only few items still need to be resolved, thus the referee suggests minor revise and acceptance after the authors addressed these issues.

Specific aspects

13 line

“water that formed the alteration”

maybe would be better „water that produced the alteration”

19 line

“role in promoting widespread carbon”

do you mean widespread carbon occurrence?

31 line

please also cite https://www.liebertpub.com/doi/full/10.1089/ast.2015.1460 for general astrobiology relevance

54 line

„present-day presence”

would sound better ”present day existence”

58 line

delete „body, being a„ as the sentence sounds better without this

60 line

“bringing water and organics”

would sound better with a „there” at the end

176 line

“paragon size-degree”

explain in bracket

275 line

“μmand”

put a space

278 line

the dimensions are wrong here, mm is written instead of micrometer

290 line

change the closing bracket from “]” to “)”

304 line

“carbonates(3.9”

put a space

305 line

“))” delete one of the double bracket

326 line

„5. Ceres mineralogy” consider to modify to „5. Overview of Ceres mineralogy”

341 line

“Ceres [20], [17].”

suggest to put the references in one bracket, need not two

Figures:

In the text at most locations the wavelength is indicated as micrometer, while in the figures occasionally as reciproc centimetre (cm-1), this should be unified somehow. The referee suggests use the simplest way: put another horizontal axis at the top of those curves with micrometer, where the horizontal axis is in cm-1. In the figures where the axis was originally micrometer no actions is necessary (as micrometer is more widely used). And where cm-1 is indicated in the text, put the corresponding value in cm-1 in bracket there.

In Figure 3b the different colour lines are hardly visible in black and white printing- This is not an issue at the other figures, but would be good to improve at this one.

Please also check

- double spaces, at some locations there seem to be double spaces instead of single ones

- check do you really not need space between the numerical values and the dimension after it

- put the numbers to lower index in the chemical formulae

Author Response

General aspects

The manuscript discusses the observations and possible origin of some organic components and other minerals identified at specific areas of the dwarf planet Ceres. It presents new results, important for the readers. The structure of the manuscript is good, its language is very good, the illustrations are relevant, the cited literature is also relevant. There are only few items still need to be resolved, thus the referee suggests minor revise and acceptance after the authors addressed these issues.

Specific aspects

13 line -> done

“water that formed the alteration”

maybe would be better „water that produced the alteration”

19 line -> done

“role in promoting widespread carbon”

do you mean widespread carbon occurrence?

31 line -> done

please also cite for general astrobiology relevance https://www.liebertpub.com/doi/full/10.1089/ast.2015.1460

54 line -> done

„present-day presence”

would sound better ”present day existence”

58 line -> done

delete „body, being a„ as the sentence sounds better without this

60 line -> done

“bringing water and organics”

would sound better with a „there” at the end

176 line -> the sentence has been clarified

“paragon size-degree”

explain in bracket

275 line -> done

“μmand”

put a space

278 line -> done

the dimensions are wrong here, mm is written instead of micrometer

290 line -> done

change the closing bracket from “]” to “)”

304 line -> done

“carbonates(3.9”

put a space

305 line -> two brackets are needed here to close the one that started with (i.e …. And the one with wavelength indication of carbonates.

“))” delete one of the double bracket

326 line -> done

„5. Ceres mineralogy” consider to modify to „5. Overview of Ceres mineralogy”

341 line -> done

“Ceres [20], [17].”

suggest to put the references in one bracket, need not two

Figures:

In the text at most locations the wavelength is indicated as micrometer, while in the figures occasionally as reciproc centimetre (cm-1), this should be unified somehow. The referee suggests use the simplest way: put another horizontal axis at the top of those curves with micrometer, where the horizontal axis is in cm-1. In the figures where the axis was originally micrometer no actions is necessary (as micrometer is more widely used). And where cm-1 is indicated in the text, put the corresponding value in cm-1 in bracket there. -> we could not find a figure with cm-1 in the manuscript, there are figures with nm rather than um but the conversion is extremely straightforward (no inversion just a division by 1000 therefore we prefer not to change the fugures).

In Figure 3b the different colour lines are hardly visible in black and white printing- This is not an issue at the other figures, but would be good to improve at this one.

Please also check

- double spaces, at some locations there seem to be double spaces instead of single ones -> done

- check do you really not need space between the numerical values and the dimension after it

- put the numbers to lower index in the chemical formulae

Reviewer 2 Report

I commend the authors on a compelling paper. I enjoyed reading it and think it is a valuable contribution to the field. Congratulations on your efforts.

Below I have noted some minor suggested changes related to formatting and phrasing. None are extensive and they should not take much time to implement.

---------
General Comments:

General Note 1:

Throughout the manuscript there are several places where extra spaces appear to be added after words.

General Note 2:

For the figures, ensure the final versions have a sufficiently high resolution. Some of the figures, especially the text, are low resolution (e.g., Figure 3 y-axis, etc.)

----------

Individual Line Comments:

L11

“Ceres composition includes” – Needs apostrophe or change to “composition of Ceres”

L12

Change to “a high quantity”

L12

Change to “The inferred mineralogy of Ceres” or similar.

L15

“to explain also the” sounds awkward – I suggest rewording.

L19

Change “played” to playing, or restructure sentence.

L24/25

“having physical and 24 chemical characteristics that are odd with the typical main belt bodies.” – awkward phrasing, not quite sure what this means.

L30

Add space in “Figure1”

L34

“Water and OH emission has been 34 detected at Ceres [7] and reference therein” unsure what this means

L35

“The” Dawn mission, or “NASA’s Dawn mission” or similar. This also applies throughout the manuscript.

L35-37

These two sentences should flow better – do you mean the most striking evidence of brine exposure?

L58-60

Awkward phrasing, I would suggest rewording. I also suggest not including subjective and non-specific qualifiers as well (i.e., “Ceres is an extremely interesting body”)

L63

Change “reference” to references

L64

I think you mean “consisting of”

L240/260/264/321/389/etc.

Sometimes “indeed” is added to the start/middle of a sentence needlessly – unless I am missing something I am not sure if you are trying to stress a point or if it’s just filler. From what I can tell it can be removed without altering the meaning of the sentences.

L256

“Likewise, also an ubiquitous” – this phrasing is awkward

L292

“as being mixed”

L316:

“However, it is very unlikely that such an unusual body –“

It seems text was deleted here – I’m not sure what this sentence is trying to say.

L318:

I would suggest using different verbiage than “we think” to make it sound less subjective and more scientific.

 L330:

A shape

L348:

I suggest a more specific word than “huge” (tall? Wide? Etc.)

L365

Unnecessary comma after faculae

L368

Again “Ceres composition” should be changed to “Ceres’ composition” or “Composition of Ceres” or similar. Check throughout manuscript for this to be consistent.

L378

Change bond to bonds

L383

Change to “are not easily identifiable”

L385-387

Many commas in this sentence – it’s hard to read. I suggest revising this.

L426-428

This sentence doesn’t tie very well to the prior sentences – I would end with a stronger sentence.

Author Response

I commend the authors on a compelling paper. I enjoyed reading it and think it is a valuable contribution to the field. Congratulations on your efforts.

Below I have noted some minor suggested changes related to formatting and phrasing. None are extensive and they should not take much time to implement.

---------
General Comments:

General Note 1:

Throughout the manuscript there are several places where extra spaces appear to be added after words.

General Note 2:

For the figures, ensure the final versions have a sufficiently high resolution. Some of the figures, especially the text, are low resolution (e.g., Figure 3 y-axis, etc.)

----------

Individual Line Comments:

L11

“Ceres composition includes” – Needs apostrophe or change to “composition of Ceres” -> done

L12 -> done

Change to “a high quantity” -> done

L12 ->done

Change to “The inferred mineralogy of Ceres” or similar. -> done

L15 -> we used a different wording

“to explain also the” sounds awkward – I suggest rewording.

L19 -> done

Change “played” to playing, or restructure sentence.

L24/25 -> we used a different wording

“having physical and 24 chemical characteristics that are odd with the typical main belt bodies.” – awkward phrasing, not quite sure what this means.

L30 -> done

Add space in “Figure1”

L34 -> we used a different wording

“Water and OH emission has been 34 detected at Ceres [7] and reference therein” unsure what this means

L35 -> done

“The” Dawn mission, or “NASA’s Dawn mission” or similar. This also applies throughout the manuscript.

L35-37 -> we used a different wording

These two sentences should flow better – do you mean the most striking evidence of brine exposure?

L58-60 -> we used a different wording

Awkward phrasing, I would suggest rewording. I also suggest not including subjective and non-specific qualifiers as well (i.e., “Ceres is an extremely interesting body”)

L63 -> done

Change “reference” to references

L64 -> done

I think you mean “consisting of”

L240/260/264/321/389/etc. -> done with one exception where we think it was needed as a logical connection between sentences.

Sometimes “indeed” is added to the start/middle of a sentence needlessly – unless I am missing something I am not sure if you are trying to stress a point or if it’s just filler. From what I can tell it can be removed without altering the meaning of the sentences.

L256 -> we used a different wording

“Likewise, also an ubiquitous” – this phrasing is awkward

L292 -> done

“as being mixed”

L316: -> we used a different wording

“However, it is very unlikely that such an unusual body –“

It seems text was deleted here – I’m not sure what this sentence is trying to say.

L318: -> we used a different wording

I would suggest using different verbiage than “we think” to make it sound less subjective and more scientific.

 L330: -> done

A shape

L348: -> we used a different wording

I suggest a more specific word than “huge” (tall? Wide? Etc.)

L365 -> done

Unnecessary comma after faculae

L368 -> done

Again “Ceres composition” should be changed to “Ceres’ composition” or “Composition of Ceres” or similar. Check throughout manuscript for this to be consistent.

L378 -> done

Change bond to bonds

L383 -> done

Change to “are not easily identifiable”

L385-387 -> we used a different wording

Many commas in this sentence – it’s hard to read. I suggest revising this.

L426-428

This sentence doesn’t tie very well to the prior sentences – I would end with a stronger sentence.